# Determination of Morphological and Quality Characteristics of Naturally Growing *Thymus kotschyanus* Boiss. & Hohen. var. *kotschyanus* Populations Around of Van/Türkiye

**DOI:** 10.3390/plants14050729

**Published:** 2025-02-27

**Authors:** Lütfi Nohutçu, Murat Tunçtürk, Rüveyde Tunçtürk, Ezelhan Şelem, Hüseyin Eroğlu

**Affiliations:** 1Department of Field Crops, Faculty of Agriculture, Van Yüzüncü Yıl University, 65090 Van, Türkiye; murattuncturk@yyu.edu.tr (M.T.); ruveydetuncturk@yyu.edu.tr (R.T.); 2Department of Landscape and Ornamental Plants, Muradiye Vocational School, Van Yüzüncü Yıl University, 65090 Van, Türkiye; ezelhanselem@yyu.edu.tr; 3Department of Biology, Faculty of Science, Van Yüzüncü Yıl University, 65090 Van, Türkiye; huseyineroglu@yyu.edu.tr

**Keywords:** essential oil, carvacrol, population, phenolic content, thyme, thymol

## Abstract

In this study, morphological and quality characteristics (nutritional value, EO ratio and content, TPC, TFC, and TAA) of 12 different naturally growing populations (T1 to T12) of *Thymus kotschyanus* var. *kotschyanus* were investigated. In the case of macro and micro nutrients, all the populations have a rich ingredient, and for heavy metal content, all population results are within limits. The percentage of essential oils in the population varied between 0.43% to 4.66% (*v*/*w*). Thymol was the most abundant compound in the whole population and the percentage of thymol ranged from 4.07% to 81.15%. In the study, eight populations had more than 50% thymol content and the maximum percentage was recorded from the T1 population (81.15%). The total phenolic compound ranged from 152.81 to 195.23 mg GAE/g of dry extract and total flavonoid content ranged from 145.24 to 382.74 mg QE/100 g. Total antioxidant activity varies between 78.43 and 228.55 µmol TE/g and the highest value was obtained from population T7. PCA analysis was carried out to determine the morphological and quality parameters of the populations. Four populations were superior to others for morphological analysis and two populations were superior to others for quality analysis. According to the result of the study, the T10 population has higher yield and quality compared to other populations.

## 1. Introduction

Türkiye, which hosts a diverse array of natural ecosystems, is home to over 1000 medicinal and aromatic plant species, some of which originated in the region [1,2]. Commercially, approximately 400 species are utilized, predominantly sourced directly from the wild [3]. However, it is noted that only a small fraction of these species are intentionally cultivated. Regrettably, factors such as indiscriminate harvesting from natural habitats and fluctuations in export demands have jeopardized the sustainability of these plant populations, thereby underscoring the imperative for cultivating these valuable species [2].

The term “thyme” collectively refers to plant species belonging to five genera (Thymus, Coridothymus, Satureja, Origanum, and Thymbra) within the Lamiaceae family [4]. The Lamiaceae family, which encompasses a diverse range of 236 genera and approximately 7200 species, is well represented in Turkey with 46 genera and 586 species, making it the third-richest family. Notably, 44.2% of these species are endemic to the region [5,6]. Furthermore, Turkey is home to 40 Thymus species, of which 18 are endemic, resulting in an endemism rate of 45% [2].

*Thymus* L. is a medicinal plant belonging to the Lamiaceae family, which is commercially important due to its aromatic and essential oil content [7,8]. This genus is known to contain approximately 270 terpenes, with thymol, linalool, carvacrol, borneol, geraniol, and p-cymene being the dominant compounds. Thymus species are a significant source of monoterpenoid phenols [9]. The unique aroma of Thymus species is primarily attributed to the presence of thymol and carvacrol, with thymol found in higher concentrations. Thymol, a crystalline and antimicrobial compound, is widely used in pharmacology, cosmetics, food flavoring, and perfumery due to its desirable properties [10,11]. Similarly, carvacrol, which exhibits antibacterial and antifungal activities, is utilized in food preservation; as an antihistamine, antioxidant, and insecticide; and for weed control [12].

*Thymus* L. species have diverse applications, serving as ingredients in pharmaceutical products, cosmetics, and as a widely used culinary spice. Previous studies have explored the utilization of Thymus species for various purposes, including their use as spices and herbal teas [13,14], as antimicrobial–antioxidant agents to extend the shelf life of foods, in medicine and pharmacy [15], as weed control agents in organic agriculture (utilizing the allelopathic effect of essential oils [16,17], and for pest control [18]). Additionally, Thymus species have found applications in organic animal husbandry as feed rations and in vegetable dyeing [19]. The specific species, *T. kotschyanus* var. *kotschyanus*, which is the focus of this research, is naturally distributed in Eastern Anatolia. The widespread use of this species in the production of herbal cheese, spices, and medicines has contributed to a decrease in its population density, highlighting the importance of conserving this natural resource. This study aimed to identify the optimal population, using wild-growing populations as the source of raw material, and to produce high-quality products through cultivation, with the goal of preserving the native flora and natural resources.

## 2. Results

The results of various morphological parameters in different populations of *T. kotschyanus* var. *kotschyanus* are presented in Table 1. Across all the studied populations, the plant height ranged from 10.0 cm to 20.0 cm. The fresh and dry weight of the plants spanned from 25.0 to 82.0 g per plant and 13.9 to 31.0 g per plant, respectively. The dry leaf weight varied between 3.40 and 13.20 g per plant, while the dry leaf ratio was 17.2% to 49.4%. In terms of the morphological observations, the T2 population exhibited the highest yield for the aerial part.

The results on the elemental composition of *T. kotschyanus* var. *kotschyanus* are presented in Table 2. The macro-elements calcium, potassium, magnesium, and sodium are expressed in g kg^−1^, while the other elements are reported in mg kg^−1^ as mean values with their absolute standard deviations based on three replicates. The calcium content among the populations ranged from 10.97 to 21.27 g kg^−1^, with the T11 population exhibiting the highest calcium level. The potassium content varied between 9.43 and 22.9 g kg^−1^, and the sodium content ranged from 26.5 to 27.2 g kg^−1^. Though minimal differences were observed in sodium content across the populations, the T2 population had the richest content for both potassium and sodium. The magnesium levels were found to be between 2.1 and 4.2 g kg^−1^.

The iron content ranged from 522.2 to 903.1 mg kg^−1^, while the manganese content varied from 38.5 to 104.0 mg kg^−1^. The T3 population exhibited the highest levels of both elements. The zinc content of the populations was found to be between 12.5 and 49.5 mg kg^−1^. The molybdenum content ranged from 0.01 to 1.36 mg kg^−1^, with the T5 population having the highest molybdenum content but the lowest zinc content among the studied populations.

The study examined populations of *T. kotschyanus* var. *kotschyanus* and found that the lead content ranged from 0.227 to 1.345 mg kg^−1^, while the arsenic content varied between 0.255 and 0.724 mg kg^−1^. The cobalt levels were observed to be between 0.140 and 0.349 mg kg^−1^, and the copper levels fluctuated from 3.39 to 7.61 mg kg^−1^. Nickel and selenium were analyzed but not detected in any population, except for the T9 population, where nickel was measured at 0.31 mg kg^−1^. (See Table 3).

The chemical composition of the essential oil from *T. kotschyanus* var. *kotschyanus* was analyzed using GC-MS, and the results are presented in Table 4. Forty-eight components were identified. The essential oil content in the populations ranged from 0.43% to 4.66%. Thymol was the predominant compound in the entire population, with percentages ranging from 4.07% to 81.15%. Notably, eight populations had more than 50% thymol content, with the maximum percentage recorded in the T1 population. Cymene was one of the major compounds in seven populations, with concentrations between 4.18% and 8.71%. Alpha-terpineol and borneol were major compounds in four populations, ranging from 4.30% to 8.0%. Carvacrol and 1,8-cineole were the principal compounds in three different populations, with ranges of 3.95–4.25% and 4.26–19.34%, respectively. Geraniol, myrcene, and bicyclogermacrene were also major compounds in two populations. Alpha-terpineol was the sole major compound for the T2 population, with a high percentage of 41.9%. Linalool was the predominant compound only in the T7 population, with a percentage of 19.16%. Germacrene was the main compound for T12, and farnesol was the main compound for T4. The analysis of the data in Table 4 indicates that an increase in thymol levels is associated with a decrease in the levels of other major compounds due to the relative quantification.

The total phenolic and flavonoid contents and antioxidant activity of *T. kotschyanus* var. *kotschyanus* populations are listed in Table 5. The total phenolic compound ranged from 152.81- to 195.23 mg GAE/g of dry extract. The population of T11 has the lowest value for all TPC, TFC, and TAA with values of 152.81 mg GAE/g, 145.24 mg QE/100 g, and 78.43 µmol TE/g, respectively. The total flavonoid content ranged from 145.24 to 382.74 mg QE/100 g and T3 exhibited a higher TFC compared with those of the other populations. The total antioxidant activity varies between 78.43 and 228.55 µmol TE/g and the highest value is obtained from population T7. This result shows that the *T. kotschyanus* var. *kotschyanus* populations are mostly rich in polyphenolic compounds. (See Figure 1).

Cluster analysis was conducted to group the different populations based on their morphological characteristics. The results of the cluster analysis, including similarities and dissimilarities, are presented in Figure 2. The dendrogram, constructed using Ward’s method, revealed two main clusters of populations. Cluster 1 was further divided into two subclusters, encompassing 11 populations of *T. kotschyanus* var. *kotschyanus*. Notably, population T2 formed a distinct cluster on its own.

## 3. Discussion

Previous studies have reported that the plant height and dry herbal weight of *T. kotschyanus* ranged from 13.6 to 18.6 cm and 33.2 to 152.1 g per plant, respectively, across five different locations [20]. Similarly, another study found the fresh and dry herbal weights to be between 14.76 and 24.78 g per plant and 4.24 and 7.10 g per plant, respectively [21]. According to prior research, the morphological characteristics of populations grown in different environments, despite belonging to the same species, can exhibit substantial variation. These findings are consistent with our own observations, considering the environmental conditions and other factors. Calcium is an essential mineral that plays a crucial role in maintaining strong bones, teeth, and blood. It also contributes to the proper functioning of muscles and nerves [22]. Additionally, calcium is required for the synthesis of the neurotransmitter acetylcholine, the activation of enzymes like pancreatic lipase, and the absorption of vitamin B from food. The recommended daily intake of calcium ranges from 500 to 800 mg for children and 80 mg for adults [23]. Potassium, on the other hand, is necessary for the activation of certain enzymes, including a co-enzyme crucial for healthy muscle growth and function [23]. Magnesium is an essential cofactor for numerous enzymes that participate in carbohydrate metabolism, and it also plays a role in insulin action, glucose regulation, and the development of type 2 diabetes. Research has shown that magnesium supplementation can enhance insulin-mediated glucose elimination and insulin production [24]. In a previous study with similar findings to ours, the elemental composition of *Thymus kotschyanus* was reported to be 11.74–15.74 mg/g calcium, 2.69–4.69 mg/g magnesium, and 8.79–18.18 mg/g potassium [8].

Iron is a critical element for human health. Up to 30% of the body’s iron is stored in the spleen, liver, and bone marrow as ferritin and hemosiderin. A smaller portion is bound to the blood transport protein transferrin. Iron is a constituent of hemoglobin, myoglobin, and numerous enzymes [25,26]. Manganese is an essential trace metal present in biological materials, but excess levels can be toxic to plants and animals. Zinc is a component of various enzymes, including alkaline phosphatase, alcohol dehydrogenase, carbonic anhydrase, and ribonucleic polymerases [26].

Heavy metals pose significant health risks, as they can adversely affect various organs and bodily systems in humans. At high exposure levels, the central nervous system, kidneys, and blood are particularly vulnerable to damage, and excessive exposure can even lead to death. Even at lower levels, heavy metals can disrupt heme synthesis and other biochemical processes, impairing psychological and neurobehavioral abilities [27,28]. The established maximum permissible limits for certain heavy metals include 1 μg kg^−1^ for lead and cadmium, 2.0 mg kg^−1^ for arsenic, 67.9 mg kg^−1^ for nickel, and 73.3 mg kg^−1^ for copper [29,30,31].

According to previous studies, the essential oil content of *T. kotschyanus* populations has been reported to range from 1% to 1.41% [32], 1.4% [33], and 1.55–1.70% [34]. The results indicate that the essential oil content of the species can vary depending on factors such as soil, environment, harvesting, drying, and analytical methods. The composition of essential oils also can vary considerably depending on multiple factors. Previous studies have reported the major essential oil components of *T. kotschyanus* var. *kotschyanus* to include thymol (31.2%), carvacrol (19.5%), and p-cymene (11.2%). Another report identified the principal constituents as thymol (48%), borneol (7.7%), 1,8 cineole (5.9%), thymol methyl ether (4.1%), β -caryophyllene (2.9%), and carvacrol (2.3%) [35].

The two most abundant compounds found in essential oils are thymol, also known as 2-isopropyl-5-methylphenol [36], and carvacrol, known as 5-isopropyl-2-methylphenol [37]. These compounds possess antibacterial, antifungal, insecticidal [38], and antioxidant properties [39], which underpin their widespread utilization in the cosmetic, food, and pharmaceutical industries [40].

The principal component analysis results revealed insights into the classification of the studied populations. The first principal component analysis, based on morphological parameters (PH, FHW, DHW, DLW, and DLR), explained 69.70% of the total variance, with 41.17% in the first dimension and 28.52% in the second dimension (Figure 3). The analysis indicated that four populations (T2, T4, T8, and T10) exhibited superior morphological characteristics. Additionally, the dry herbal weight and dry leaf rate parameters were found to have a strong positive relationship, as indicated by their narrow angle of approximately 90 degrees and distance from the central axis. The second principal component analysis, focusing on quality parameters (TPC, TFC, TAA, and EOR), explained 74.97% of the total variance, with 44.62% in the first dimension and 30.35% in the second dimension. This analysis identified two populations (T7 and T10) with higher-quality results. Furthermore, the total phenolic compound and total antioxidant activity parameters were observed to have a strong positive relationship. Importantly, the T10 population was found to be superior to the other populations in both principal component analyses, indicating that it possessed the highest yield and the best quality characteristics among the studied samples.

## 4. Materials and Methods

### 4.1. Material

The locations of *T. kotschyanus* var. *kotschyanus* populations growing naturally in the region were determined based on flora studies. Samples of *T. kotschyanus* var. *kotschyanus* were collected from 13 different sites in Van, Turkey, during May 2020 when the plants were in the early flowering stage. The geographical features of the collection sites are provided in Table 6. Three individual plants were selected and harvested from each location. The plants were photographed in their natural habitats, and necessary macroscopic measurements were taken. The plants were then identified as herbarium specimens through microscopic measurements and observations conducted in the Herbarium of the Department of Biology, Faculty of Science, Van Yüzüncü Yıl University. “Flora of Turkey” was the primary reference used for the identification of these specimens, with the other auxiliary sources utilized when necessary.

### 4.2. Methodology

#### 4.2.1. Morphological Measurements

In the morphological analysis, five quantitative features were examined from the vegetative organs of *T. kotschyanus* var. *kotschyanus* plants. Prior to harvesting, the plant height was measured from the soil surface to the apex. The fresh herbal weight was determined by harvesting and weighing the aerial parts of the plant, 5 cm above the soil surface, and recorded in grams per plant. These plant height and fresh herbal weight parameters were measured in the field. Subsequently, the collected samples were washed with deionized water and dried at room temperature for 2 days, and the dry herbal weight was measured in grams per plant. The stem and leaf of the samples were separated and weighed, and the rate of dry leaf and dry leaf yield was calculated.

#### 4.2.2. Quality Features

The dried leaf of the plant was put into the distillation unit along with 1200 mL water and the oil was separated by the hydrodistillation method for 3 h using a Clevenger-type apparatus. The essential oil was extracted and measured, and the rate of essential oil was determined (*v*/*w*). Each essential oil sample was analyzed by GC-MS (Gas Chromatography–Mass Spectrometry (Agilent 7890B GC/5977E-USA)) to determine the essential oil profile of the sample. The analyses were carried out using a TRB-Wax MS model, and a 5% Phenyl Polysilphenylene-siloxane, 0.25 mm × 30 m, film thickness 0.25 μm capillary column was employed. The ionization energy was 70 eV and the mass range was *m*/*z* 1.2–1200 amu. The oven temperature was programmed with different stationary phases: starting with 60 °C, then increased by 3 °C/min to 240 °C. The constituents of the essential oils were determined by matching their relative retention times and mass spectra with real samples from the essential oil library data (Nist 27, Wiley, 7, and Nist 147) and comparing their relative retention indices (RRIs) with published data.

For mineral analysis, the plant samples were reduced to ashes in a furnace by nitric acid (AR) and hydrochloric acid [41], and then distilled water (50 mL) was added to the samples in a volumetric flask. All the analysis was repeated 3 times to achieve accuracy. Mineral contents were determined by atomic absorption spectrometry (AAS (Thermo Scientific/Waltham, MA, USA)) and Inductively Coupled Plasma-Optical Emission Spectroscopy (ICP-OES (Thermo Scientific/USA)). Average data were calculated by computer office programs and given with standard deviations.

The total phenolic compound content was measured according to the Obanda–Owuor method [42]. The antioxidant activity was also performed based on the antioxidant power (FRAP) (Iron (III) antioxidant power reduction) method [43] followed by readings of the absorbance at 593 nm, and antioxidant activity values were recorded as Trolox equivalent (TE)/mg. The total flavonoid content was determined with some modifications according to the method developed by Quettier-Deleu et al. [44]. The total amount of flavonoid was measured at 415 nm and calculated in mg quercetin equivalent (QE) 100 g^−1^ DM by using the calibration curve prepared using standard quercetin.

## 5. Conclusions

The present study aimed to evaluate the morphological characteristics and chemical composition of twelve populations of *T. kotschyanus* var. *kotschyanus*. The results revealed that some populations exhibited a notably high ratio of essential oils which were typically rich in thymol. Additionally, the study demonstrated that the *T. kotschyanus* var. *kotschyanus* populations are nutritionally abundant in terms of macro and micro elements. Furthermore, most of the populations exhibited high antioxidant activity, total phenolic content, and flavonoid content, rendering them suitable for consumption as food, flavoring, or spice.

The study found that some *Thymus kotschyanus* var. *kotschyanus* populations exhibited superior morphological and quality traits. Specifically, the populations T2, T4, T8, and T10 were identified as having superior morphological characteristics compared to the other tested populations. Additionally, the T7 and T10 populations demonstrated higher quality attributes. Based on the findings of this study, the T10 population exhibited the most favorable overall performance, indicating that it would be the primary candidate for selection and cultivation. Furthermore, following a thorough field trial, the researchers recommend this population for further investigation.

## Figures and Tables

**Figure 1 plants-14-00729-f001:**
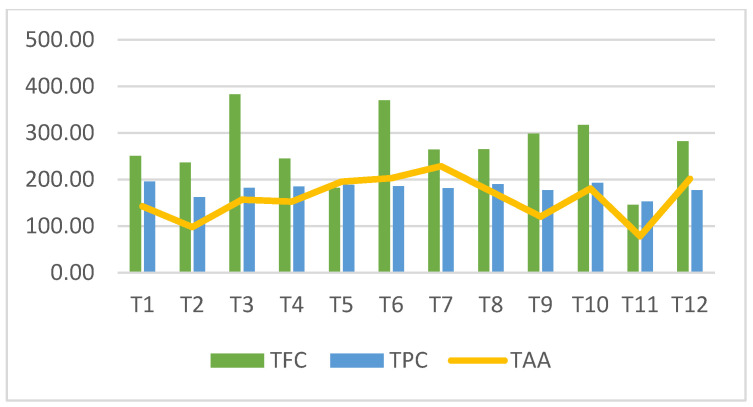
Total flavonoid content (TFC), total phenolic content (TPC), and total antioxidant activity (TAA) of *T. kotschyanus* var. *kotschyanus* populations (T1–T12 represent the examined populations of study).

**Figure 2 plants-14-00729-f002:**
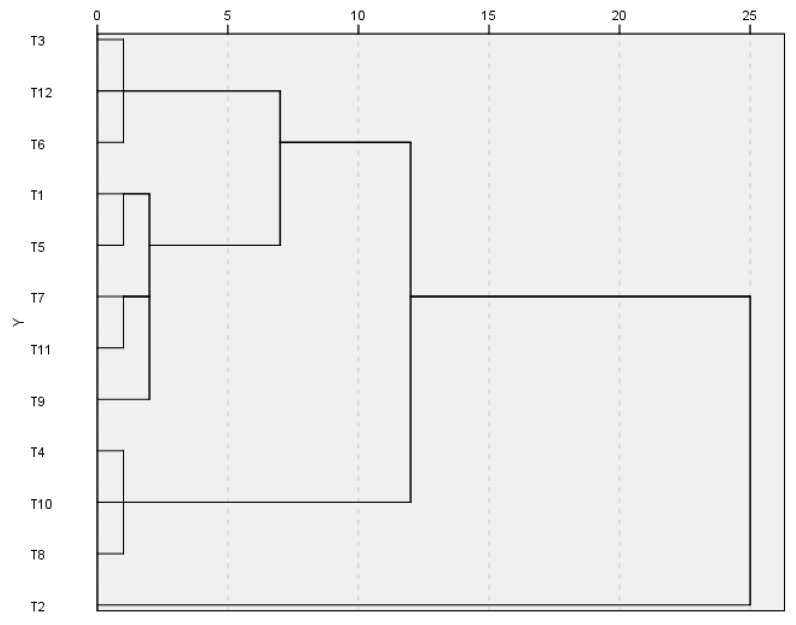
Grouping dendrogram of *T. kotschyanus* var. *kotschyanus* populations using Ward’s method (T1–T12 represent the examined populations of study).

**Figure 3 plants-14-00729-f003:**
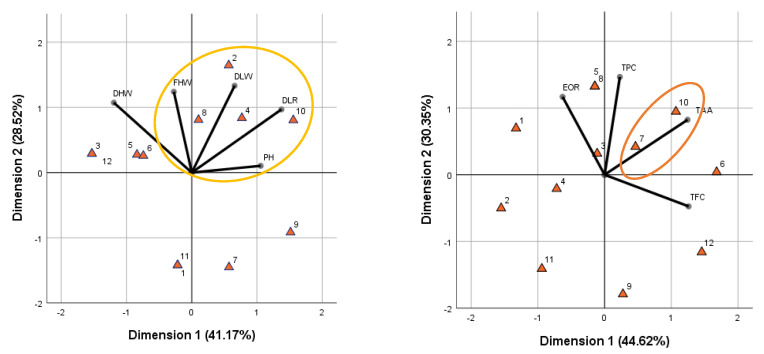
PCA analysis for classification of studied morphological and quality parameters of populations (PH, plant height; FHW, fresh herbal weight; DHW, dry herbal weight; DLW, dry leaf weight; DLR, dry leaf ratio; TPC, total phenolic compound; TFC, total flavonoid compound; TAA, total antioxidant activity; EOR, essential oil ratio; 1–12 represent the examined populations of study).

**Table 1 plants-14-00729-t001:** Morphological measurement of *T. kotschyanus* var. *kotschyanus* population with standard deviation.

P. No	PH (cm)	FHW (g Plant^−1^)	DHW (g Plant^−1^)	DLW (g Plant^−1^)	DLR (%)
T1	11.30 ± 2.08 i	38.62 ± 10.26 g	16.44 ± 5.38 j	5.22 ± 2.60 g	31.92 ± 1.13 d
T2	20.02 ± 10.44 a	82.06 ± 18.57 a	31.00 ± 5.53 a	13.24 ± 0.34 a	43.54 ± 6.80 b
T3	12.00 ± 3.00 h	45.32 ± 11.01 d	21.90 ± 6.48 d	3.81 ± 1.44 k	17.20 ± 1.40 g
T4	14.08 ± 2.64 f	46.60 ± 11.37 c	18.20 ± 2.03 h	7.47 ± 2.44 e	40.38 ± 9.19 c
T5	13.60 ± 2.51 g	36.68 ± 12.05 h	23.10 ± 10.32 c	8.00 ± 3.55 d	32.04 ± 3.72 d
T6	19.30 ± 1.52 b	50.05 ± 12.00 b	24.23 ± 11.18 b	6.55 ± 2.73 f	24.70 ± 14.18 e
T7	15.01 ± 2.64 e	34.04 ± 7.08 i	15.94 ± 7.44 k	3.46 ± 2.41 l	23.42 ± 7.28 ef
T8	10.60 ± 1.15 j	43.30 ± 8.32 f	20.50 ± 5.84 f	10.05 ± 0.70 c	49.48 ± 4.10 a
T9	17.66 ± 6.65 c	25.00 ± 3.00 k	13.98 ± 0.74 l	4.53 ± 0.77 i	32.41 ± 4.61 d
T10	17.00 ± 1.73 d	44.62 ± 6.10 e	18.05 ± 10.90 i	11.27 ± 3.55 b	43.60 ± 4.63 b
T11	10.32 ± 1.52 k	28.64 ± 4.16 j	19.50 ± 2.09 g	4.65 ± 1.15 h	23.93 ± 4.70 ef
T12	10.01 ± 2.00 l	46.60 ± 6.63 c	20.80 ± 12.99 e	4.22 ± 1.89 j	21.75 ± 8.03 f
CV	4.28 **	9.65 **	3.51 **	7.11 **	3.37 **

T1–T12 represent the examined populations of study; PL—plant height; FHW—fresh herbal weight; DHW—dry herbal weight; DLW—dry leaf weight; DLR—dry leaf ratio; CV—coefficient of variation; The data in columns with the same letters were not significantly different from each other, based on the Duncan multiple range test (DMRT); ** significant at *p* < 0.01.

**Table 2 plants-14-00729-t002:** Nutritional content of *T. kotschyanus* var. *kotschyanus* population with standard deviation.

P. No	Ca (g kg^−1^)	K (g kg^−1^)	Mg (g kg^−1^)	Na (g kg^−1^)	Fe (mg kg^−1^)	Mn (mg kg^−1^)	Mo (mg kg^−1^)	Zn (mg kg^−1^)
T1	16.17 ± 0.71 b	13.98 ± 0.44 e	2.44 ± 0.20 h	26.86 ± 0.46 f	558.03 ± 17.11 k	55.93 ± 1.60 e	0.52 ± 0.09 e	25.48 ± 2.70 e
T2	11.52 ± 0.31 h	22.89 ± 3.02 a	2.30 ± 0.09 i	27.27 ± 0.53 a	805.5 ± 31.21 c	64.01 ± 2.93 c	0.72 ± 0.25 d	31.18 ± 2.60 c
T3	11.17 ± 1.48 i	14.45 ± 3.02 a	2.96 ± 0.39 g	26.88 ± 0.58 e	903.19 ± 97.19 a	104.07 ± 3.72 a	0.11 ± 0.05 i	28.99 ± 2.88 d
T4	13.62 ± 1.70 f	14.37 ± 1.03 d	2.21 ± 0.15 j	26.79 ± 0.47 h	591.05 ± 56.74 i	49.03 ± 12.23 i	0.19 ± 0.04 h	24.55 ± 1.09 fg
T5	13.94 ± 0.65 e	10.34 ± 1.52 h	3.75 ± 1.41 c	26.69 ± 0.45 i	522.27 ± 77.23 l	55.16 ± 0.96 e	1.36 ± 0.02 a	12.57 ± 0.22 i
T6	16.32 ± 0.31 b	15.17 ± 0.48 c	3.29 ± 1.17 e	26.61 ± 0.46 j	693.39 ± 29.59 g	67.92 ± 3.85 b	0.93 ± 0.03 c	49.54 ± 8.35 a
T7	14.59 ± 2.24 d	13.25 ± 0.23 f	3.82 ± 0.84 b	27.07 ± 0.46 b	712.73 ± 70.19 f	54.18 ± 11.85 f	0.28 ± 0.20 g	33.41 ± 3.15 b
T8	10.97 ± 0.40 j	12.34 ± 1.49 g	2.15 ± 0.14 k	27.02 ± 0.46 c	626.76 ± 57.89 h	45.20 ± 13.21 j	0.01 ± 0.01 j	14.46 ± 2.40 h
T9	15.24 ± 1.67 c	10.11 ± 0.92 i	3.34 ± 0.06 d	26.52 ± 0.60 k	885.33 ± 65.62 b	53.02 ± 5.09 g	1.15 ± 0.16 b	23.97 ± 1.43 g
T10	11.86 ± 0.95 g	16.01 ± 1.04 b	2.46 ± 0.23 h	26.99 ± 0.46 d	575.44 ± 81.98 j	62.19 ± 1.20 d	0.34 ± 0.14 f	24.89 ± 1.50 f
T11	21.27 ± 2.40 a	9.42 ± 0.61 j	3.08 ± 0.52 f	26.83 ± 0.50 g	797.15 ± 105.86 d	38.50 ± 6.05 k	0.18 ± 0.01 h	24.13 ± 2.72 g
T12	11.85 ± 0.53 g	13.13 ± 1.11 f	4.21 ± 1.28 a	26.49 ± 0.48 l	737.52 ± 64.57 e	50.94 ± 8.99 h	0.19 ± 0.12 h	31.06 ± 3.71 c
CV	0.63 **	0.76 **	0.66 **	0.02 **	0.55 **	0.84 **	2.84 **	1.05 **

T1–T12 represent the examined populations of study; CV—coefficient of variation; The data in columns with the same letters were not significantly different from each other, based on the Duncan multiple range test (DMRT); ** significant at *p* < 0.01.

**Table 3 plants-14-00729-t003:** Heavy metal content of *T. kotschyanus* var. *kotschyanus* population with standard deviation.

P. No	As (mg kg^−1^)	Co (mg kg^−1^)	Cu (mg kg^−1^)	Ni (mg kg^−1^)	Pb (mg kg^−1^)
T1	0.53 ± 0.12 c	0.19 ± 0.01 e	4.91 ± 1.51 f	n.d.	0.42 ± 0.19 h
T2	0.54 ± 0.20 b	0.19 ± 0.04 e	7.29 ± 2.73 b	n.d.	0.22 ± 0.13 l
T3	0.47 ± 0.03 e	0.34 ± 0.10 a	6.57 ± 0.38 d	n.d.	0.46 ± 0.21 g
T4	0.50 ± 0.32 d	0.14 ± 0.01 h	3.99 ± 0.33 i	n.d.	0.39 ± 0.21 i
T5	0.30 ± 0.02 g	0.17 ± 0.01 f	6.44 ± 0.30 e	n.d.	0.34 ± 0.18 j
T6	0.54 ± 0.01 b	0.19 ± 0.02 e	7.61 ± 2.62 a	n.d.	0.94 ± 0.15 c
T7	0.52 ± 0.16 c	0.15 ± 0.01 g	3.72 ± 0.54 j	n.d.	0.29 ± 0.09 k
T8	0.36 ± 0.16 f	0.17 ± 0.02 f	3.39 ± 0.60 k	n.d.	0.50 ± 0.12 f
T9	0.48 ± 0.08 e	0.34 ± 0.07 b	4.29 ± 0.72 h	0.31 ± 0.02	1.34 ± 0.51 a
T10	0.25 ± 0.09 i	0.13 ± 0.05 h	6.79 ± 2.40 c	n.d.	0.66 ± 0.02
T11	0.72 ± 0.18 a	0.22 ± 0.09 c	3.44 ± 0.35 k	n.d.	1.15 ± 0.39 b
T12	0.27 ± 0.02	0.20 ± 0.10 d	4.55 ± 0.62 g	n.d.	0.55 ± 0.07 e
CV	1.35 **	1.77 **	0.89 **		1.76 **

T1–T12 represent the examined populations of study; CV—coefficient of variation; n.d—not detected; The data in columns with the same letters were not significantly different from each other, based on the Duncan multiple range test (DMRT); ** significant at *p* < 0.01.

**Table 4 plants-14-00729-t004:** Essential oil ratio and composition of *T. kotschyanus* var. *kotschyanus* population with standard deviation.

		Compounds	Populations											
			T1	T2	T3	T4	T5	T6	T7	T8	T9	T10	T11	T12
R.I.	R.T.	Essential Oil Rate (%)	4.18	2.77	2.48	1.28	2.87	1.74	2.26	2.34	1.85	4.66	0.66	0.43
1023	6.327	Alpha pinene	0.477	0.852	3.307	1.112	2.406	1.849	0.987	1.025	1.974	1.540	-	-
1065	7.262	Camphene	0.164	0.23	0.251	0.51	0.365	0.316	0.809	0.523	0.770	0.902	-	-
1108	8.404	Beta pinene	-	0.273	-	1.561	-	-	0.488	0.211	-	0.290	-	-
1116	8.761	Sabinene	-	0.69	-	1.567	0.334	-	0.255	-	-	-	-	-
1149	10.214	Myrcene	0.443	7.344	0.755	10.147	0.418	0.498	0.4	0.355	0.273	0.831	-	-
1171	11.146	Alpha-terpinene	0.592	0.2399	0.651	-	0.354	0.369	0.291	0.781	-	1.028	-	-
1193	12.134	Limonene	-	0.43	0.529	0.667	0.35	0.389	-	-	0.399	0.245	-	-
1207	12.982	1,8-cineole	0.895	1.678	1.811	19.346	4.26	3.707	7.77	1.613	1.711	1.877	7.909	-
1236	15.165	Gamma terpinene	1.325	2.76	2.262	-	0.528	2.496	1.831	0.554	0.516	5.267	-	-
1240	15.489	Trans ocimene	-	-	-	3.981	-	-	-	-	-	-	-	-
1245	15.849	3-octanone	-	-	0.674	-	0.903	0.417	0.902	-	0.351	1.188	-	-
1258	16.845	Cymene	5.732	3.286	4.185	-	4.982	4.94	4.26	8.712	2.356	5.315	1.964	-
1440	30.837	1-octen-3-ol	-	0.288	0.38	-	-	-	0.321	-	-	0.187	-	-
1455	31.939	Trans sabinene hydrate	0.748	0.428	-	1.235	0.814	2.195	0.744	1.008	0.762	0.877	0.855	-
1491	34.568	Camphor	0.365	-	-	0.908	0.559	1.66	0.864	0.363	3.107	-	0.719	-
1533	37.409	Linalool	-	0.214	-	0.41	-	-	19.163	-	0.181	-	-	-
1535	37.557	Cis sabinene hydrate	-	-	-	-	-	-	-	0.333	-	0.295	-	-
1557	39.061	Bornyl acetate	-	-	-	-	-	-	0.451	-	0.468	-	-	-
1567	39.737	Thymyl methyl ether	0.532	0.306	-	-	-	-	-	-	-	-	-	-
1572	40.065	Caryophyllene	-	1.526	0.858	4.924	0.651	3.946	1.688	1.122	0.857	0.944	1.213	11.721
1577	40.389	Carvacrol methyl ether	0.373	1.182	2.16	-	1.487	3.197	2.012	0.704	2.710	-	-	-
1581	40.677	4-terpineol	-	-	-	0.822	0.529	0.953	0.746	0.695	-	0.488	-	-
1619	43.045	Pulegone	-	-	1.84	-	-	-	-	-	-	-	-	1.965
1630	43.726	Pinocarveol	-	-	-	-	-	-	-	-	0.308	-	-	-
1654	45.192	Carveol	-	-	-	-	-	-	-	-	-	-	0.718	-
1657	45.372	Verbenol	-	-	-	-	-	-	-	-	0.826	-		-
1662	45.660	Gamma muurolene	-	-	-	-	-	-	-	-	-	-	2.553	-
1675	46.462	Alpha-terpineol + Borneol	-	-	1.301	4.306	1.999	2.633	7.164	2.481	8.002	-	6.761	-
1676	46.533	Borneol	0.845	-	-	-	-	-	-	-	-	4.355	-	-
1680	46.764	Germacrene	-	1.455	-	1.818	-	-	-	-	-	-	-	12.415
1691	48.111	Alpha-terpineol	-	41.392	-	1.055	-	-	0.395	-	0.887	-	-	-
1701	48.008	Beta bisabolene	1.919	0.287	1.534	-	0.896	1.729	0.543	0.858	0.904	0.679	0.713	-
1704	48.171	Geranial											1.631	-
1730	49.626	Delta cadinene	-	-	0.956	-	0.642	0.83	-	0.404	-	0.519	-	-
1730	49.654	Geranyl acetate	-	-	-	-	-	-	0.473	-	2.775	-	-	-
1744	51.617	Bicyclogermacrene	-	-	-	6.109	-	-	-	-	-	-	-	7.672
1777	52.243	Nerol	-	-	-	-	-	-	-	-	0.631	-	1.796	-
1825	54.850	Geraniol	-	0.581	-	-	-	-	2.157	-	16.833	-	12.304	-
1953	61.449	Caryophyllene oxide	-	-	-	7.155	0.41	1.166	0.74	0.358	0.514	0.201	1.051	3.189
2020	64.730	Beta copaene	-	-	-	-	-	-	-	-	-	-	3.162	7.010
2028	65.128	Cubedol	-	-	-	-	0.554	-	-	-	-	-	-	-
2094	68.255	Spathulenol	-	-	0.322	3.179	-	0.608	0.436	-	0.401	-	-	4.017
2138	70.396	Tau cadinol	-	-	-	-	1.559	-	-	-	-	-	-	-
2147	70.838	**Thymol**	**81.157**	**32.946**	**70.933**	**4.073**	**69.171**	**62.95**	**43.556**	**73.946**	**50.114**	**69.625**	**49.652**	**32.634**
2176	72.282	Carvacrol	4.012	0.707	4.251	-	2.475	0.494	0.554	3.952	0.343	3.345	2.981	2.055
2200	73.432	Alpha cadinol	-	-	-	-	0.45	-	-	-	-	-	-	4.266
2327	79.425	Ledene oxide											-	1.678
2330	79.544	Farnesol	-	0.746	0.415	22.698	-	1.323	-	-	-	-	-	5.481
		Others	0.421	-	0.627	2.132	2.903	1.333	-	-	-	-	2.873	5.897

T1–T12 represent the examined populations of study; R.I.—retention indices; R.T.—retention time.

**Table 5 plants-14-00729-t005:** Total flavonoid content, total phenolic content, and total antioxidant activity of *T. kotschyanus* var. *kotschyanus* populations with standard deviation.

Population No	TFC (mg QE/100 g)	TPC (mg GAE/g)	TAA (µmol TE/g)
T1	250.27 ± 46.24 h	195.23 ± 13.58 a	142.52 ± 17.87 h
T2	236.04 ± 28.48 j	161.79 ± 29.94 k	97.97 ± 28.26 j
T3	382.74 ± 51.75 a	181.95 ± 6.29 g	156.88 ± 12.11 f
T4	245.06 ± 8.83 i	185.07 ± 3.20 f	152.29 ± 35.83 g
T5	181.87 ± 8.34 k	188.12 ± 7.07 d	195.25 ± 49.65 c
T6	369.89 ± 71.93 b	185.31 ± 3.09 e	202.63 ± 16.07 b
T7	264.16 ± 3.92 g	181.09 ± 9.50 h	228.54 ± 20.57 a
T8	264.51 ± 2.94 f	189.60 ± 9.17 c	174.44 ± 21.83 e
T9	298.54 ± 71.20 d	176.56 ± 5.30 j	120.25 ± 59.62 i
T10	317.11 ± 3.68 c	192.26 ± 3.20 b	180.59 ± 40.49 d
T11	145.24 ± 54.75 l	152.81 ± 39.12 l	78.43 ± 32.62 k
T12	281.87 ± 28.65 e	176.71 ± 7.51 i	201.50 ± 51.10 b
CV	3.08 **	1.15 **	1.27 **

T1–T12 represent the examined populations of study; TFC—total flavonoid content; TPC—total phenolic content; TAA—total antioxidant activity; QE—quercetin equivalent; TE—trolox equivalent; GAE—gallic acid equivalent; The data in columns with the same letters were not significantly different from each other, based on the Duncan multiple range test (DMRT); ** significant at *p* < 0.01.

**Table 6 plants-14-00729-t006:** Geographical feature of localities.

	District			Location		
P. No	City	County	Region	Latitude	Longitude	Altitude
T1	Van/Turkey	Saray	City center	38°39′57″ N	44°08′57″ E	2061 m
T2	Van/Turkey	Erciş	Şehir pazarı village	39°13′58″ N	43°25′09″ E	2240 m
T3	Van/Turkey	Gevaş	Kuskunkıran mountain	38°22′34″ N	42°47′11″ E	2255 m
T4	Van/Turkey	Edremit	Ayazpınar village	38°22′31″ N	43°18′29″ E	2073 m
T5	Van/Turkey	Gürpınar	Dönemeç Village	38°19′35″ N	43°14′16″ E	1705 m
T6	Van/Turkey	Gevaş	Altınsaç village	38°24′20″ N	42°53′32″ E	1674 m
T7	Van/Turkey	Tuşba	Ağartı village	38°42′26″ N	43°12′30″ E	1791 m
T8	Van/Turkey	İpekyolu	Kevenli village	38°28′08″ N	43°27′53″ E	1981 m
T9	Van/Turkey	Tuşba	Toprakkale village	38°31′24″ N	43°23′35″ E	1972 m
T10	Van/Turkey	Muradiye	City center	38°58′40″ N	43°43′33″ E	1679 m
T11	Van/Turkey	Çatak	Yukarı Narlıca village	38°07′57″ N	43°01′46″ E	2263 m
T12	Van/Turkey	Bahçesaray	Karapet passage	38°09′19″ N	42°52′50″ E	3085 m

## Data Availability

The original contributions presented in this study are included in the article. Further inquiries can be directed to the corresponding authors.

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
