# Peer review of "Determination of Morphological and Quality Characteristics of Naturally Growing Thymus kotschyanus Boiss. & Hohen. var. kotschyanus Populations Around of Van/Türkiye"

_plants, 2025, doi:10.3390/plants14050729_

Round 1
Reviewer 1 Report
Comments and Suggestions for Authors
The article is well written, but the literature already reports on the chemical composition of kotschyanus, proof of which lies in the fact that most of the articles cited are more than five years old. I believe that correlation/PCA evaluation of more variables used in the study will enrich the discussion of the work.
Line 64 - please explain "The widespread use of this species in the production of herbal cheese, spices, and medicines has contributed to an increase in its population density, highlighting the importance of conserving this natural resource."
Statistics are needed in tables 1 and 2 to say which values are higher or lower
Table 2 and 3 - has a leaf extract been made?
Line 110 - "constituting approximately 99.0-100.0% of the total oil content" - how can you be sure, as this is not an absolute quantitative analysis? Maybe you can say "100% of the profile obtained by GCMS"
Line 122 – “The analysis of the data in Table 4 suggests a negative correlation between thymol content and other major compounds” it seems to me that this is because it is a relative quantification, rephrase
Statistics are needed in table 5 to say which values are higher or lower
Line 158-172 may be part of the discussion
Line 207 - Is there an intake limit for cobalt and lead?
Lines 212-217 - what health benefits do these compounds bring? Are they known antioxidants in the literature? What is the chemical classification of these compounds?
It would be interesting to do PCA with Total phenolic compound TFC; Total flavonoid compound TAA; Total Antioxidant activity EOR; Essential Oil Ratio, and Thymol, and discuss this possible relationship. This could also occur with all other compounds.
Author Response
Dear Reviewer, Thank you for spending your valuable time and effort to review our paper and also for your constructive comments, we have carefully revised the whole work with your guidance and comments.
Comments-
The article is well written, but the literature already reports on the chemical composition of kotschyanus, proof of which lies in the fact that most of the articles cited are more than five years old. I believe that correlation/PCA evaluation of more variables used in the study will enrich the discussion of the work.
Line 64 - please explain "The widespread use of this species in the production of herbal cheese, spices, and medicines has contributed to an increase in its population density, highlighting the importance of conserving this natural resource."
Statistics are needed in tables 1 and 2 to say which values are higher or lower
Table 2 and 3 - has a leaf extract been made?
Line 110 - "constituting approximately 99.0-100.0% of the total oil content" - how can you be sure, as this is not an absolute quantitative analysis? Maybe you can say "100% of the profile obtained by GCMS"
Line 122 – “The analysis of the data in Table 4 suggests a negative correlation between thymol content and other major compounds” it seems to me that this is because it is a relative quantification, rephrase
Statistics are needed in table 5 to say which values are higher or lower
Line 158-172 may be part of the discussion
Line 207 - Is there an intake limit for cobalt and lead?
Lines 212-217 - what health benefits do these compounds bring? Are they known antioxidants in the literature? What is the chemical classification of these compounds?
It would be interesting to do PCA with Total phenolic compound TFC; Total flavonoid compound TAA; Total Antioxidant activity EOR; Essential Oil Ratio, and Thymol, and discuss this possible relationship. This could also occur with all other compounds.
Response to Reviewer
Line 64- increase written by mistake and its changed to decrease.
Statical analyzes and grouping added Table 1,2 and 5.
In study we use wet digestion method for mineral analysis. So leaf exract not needed.
Line 110- İn case of misunderstanding sentence corrected.
Line 122- The sentence rephrased.
Line 158-172 paragraph moved to discussion section.
Line 207- The limit of lead already mentioned in sentence. Cobalt is not clear as other heavy metals, WHO doesn’t have a report about it. That’s why we didn’t mention it.
Line 212-217 Information about properties of compounds and also chemical classification added.
İf possible, we would like to publish our results as Figure 3. İn the case of phenolic compound.
Reviewer 2 Report
Comments and Suggestions for Authors
Manuscript entitled "Determination of morphological and quality characteristics of naturally growing Thymus kotschyanus Boiss. & Hohen. var. kotschyanus populations around of Van/Türkiye" submitted to Plants journal is well written and the results are presented in a logical and coherent manner. The paper is adequately organized and the topic focused on the study on properties of wild population of thymus kotschyanus.
Although the manuscript is well-edited, however small improvements should be introduced/explanations that will improve its quality:
- The authors did not really present the purpose of their research. Why did they conduct such a detailed analysis of the composition of oils from a wild population? Do they intend to conduct selection for cultivation? Do they want to indicate these wild populations as the source of the raw material? The overarching purpose of the research should be explained in the text.
- Lines 24-26 All populations mentioned should have a number provided in the abstract as well
- Lines 31-32 There should be: .. over 1000 SPECIES of medicinal and aromatic plants ... not pieces, right?
- Table 1. (and Tables 2-5 and Figures 1 and 2) should contain explanations of symbols T1-T12
- Lines 69-75 and 175-180 - The authors did not attempt to explain the reason for such large differences in the mass of the plants. What could have caused this? The smallest plants yielded only 14 g of dry herb (T9) and the largest 31 g (T2) - more than double that! It is necessary to explain why there is such a difference in the size of plants of the same species.
- line 230 These photographs should be included with the manuscript, at least as supplementary material.
- line 250 - There is a certain inaccuracy. The authors write earlier (line 229) that they collected three plants from each location - one plant gives 3-13 g of leaves (Table 1) - where did 500 g of material for distillation come from?
The first paragraph of the conclusions is more like a summary. The research objective should be stated at the end of the Introduction or next to the research hypotheses (which are also not there). The conclusions should consider whether the research objective has been achieved and the hypotheses confirmed or rejected.
Author Response
Dear Reviewer, Thank you for spending your valuable time and effort to review our paper and also for your constructive comments, we have carefully revised the whole work with your guidance and comments.
Comments-
Manuscript entitled "Determination of morphological and quality characteristics of naturally growing Thymus kotschyanus Boiss. & Hohen. var. kotschyanus populations around of Van/Türkiye" submitted to Plants journal is well written and the results are presented in a logical and coherent manner. The paper is adequately organized and the topic focused on the study on properties of wild population of thymus kotschyanus.
Although the manuscript is well-edited, however small improvements should be introduced/explanations that will improve its quality:
- The authors did not really present the purpose of their research. Why did they conduct such a detailed analysis of the composition of oils from a wild population? Do they intend to conduct selection for cultivation? Do they want to indicate these wild populations as the source of the raw material? The overarching purpose of the research should be explained in the text.
- Lines 24-26 All populations mentioned should have a number provided in the abstract as well
- Lines 31-32 There should be: .. over 1000 SPECIES of medicinal and aromatic plants ... not pieces, right?
- Table 1. (and Tables 2-5 and Figures 1 and 2) should contain explanations of symbols T1-T12
- Lines 69-75 and 175-180 - The authors did not attempt to explain the reason for such large differences in the mass of the plants. What could have caused this? The smallest plants yielded only 14 g of dry herb (T9) and the largest 31 g (T2) - more than double that! It is necessary to explain why there is such a difference in the size of plants of the same species.
- line 230 These photographs should be included with the manuscript, at least as supplementary material.
- line 250 - There is a certain inaccuracy. The authors write earlier (line 229) that they collected three plants from each location - one plant gives 3-13 g of leaves (Table 1) - where did 500 g of material for distillation come from?
The first paragraph of the conclusions is more like a summary. The research objective should be stated at the end of the Introduction or next to the research hypotheses (which are also not there). The conclusions should consider whether the research objective has been achieved and the hypotheses confirmed or rejected.
Response to reviewer
Explanation added about purpose of the study.
Lines 24-26- Populations coded with T1 to T12 and added shortly to abstract.
Lines 31-32- Over 1000 plant species, its corrected.
Explanation added to all tables and figures.
Line 69-75 Differences between populations could be vary depend on climate and other environmental conditions. As reported in references 19 and line 167 on discussion section, dry herbal weight varied 33.2-152.1 g per plant in previous study, although same species highest yield almost 5 time more than lowest one. This study shows that differences of plants yield which grown on different environment, could be vary wide span. To avoid to misunderstand it mentioned in discussion section.
Line 230- there are more than 10 photographs for each location, it means about more than 100 photo, that’s why we avoid to share these photos in study.
Line 250- Its written mistakenly, it should be ± 50.0 gram but in case of misunderstand its removed.
The conclusions revised in the direction of the recommendations. Introduction also revised and aim of the study added.
Round 2
Reviewer 1 Report
Comments and Suggestions for Authors
The authors can improve the quality of the paper by including thymol in the PCA analysis. However, all other considerations have been addressed.
Comments on the Quality of English LanguageNo comments.
Author Response
Comments
The authors can improve the quality of the paper by including thymol in the PCA analysis.
However, all other considerations have been addressed.
Response to reviewer
Thank you for your contribution, The PCA analysis with thymol content can improve the quality of paper, however I have to add another PCA figure for thymol content analysis, if possible we would like to publish without it. Best Regards...